# A Comprehensive Approach of Exploring Usability Problems in Enterprise Resource Planning Systems

**Amna Asif \*, Deemah AlFrraj and Majed A. Alshamari**

Information Systems Department, College of Computer Sciences and Information Technology (CCSIT), King Faisal University, P.O. Box 400, Al-Ahsa 31982, Saudi Arabia; 218006145@student.kfu.edu.sa (D.A.); smajed@kfu.edu.sa (M.A.A.)

\*  Correspondence: aarkhan@kfu.edu.sa

**Abstract:** Enterprise Resource Planning (ERP) is a frequently used system among organizations to automate their workflows, and companies' performances are highly dependent on the ERP system. The usability issues of ERP systems may cause performance degradation, resulting in the company's loss in terms of cost. Previously, several studies reported many usability problems of ERP systems. It can be helpful for the developers and designers of ERP systems to use design recommendations as a quick reference to avoid recurrent usability problems of ERP systems. Currently, this area lacks effective consolidation of the previously reported usability problems data. This paper presents a unique approach to developing a precise checklist of ERP usability problems using the topic modeling technique. Our analysis found six different usability problem-related topics that can be generalized for various ERP systems. We have successfully validated our checklist in three different usability studies of ERP systems. The most found usability problems are "difficulty searching and finding desired item/information in interface and error handling" and "missing data and information". The outcome of our paper is the provision of recommendations to avoid the usability problems of ERP systems and help organizations efficiently prevent frequent issues during the development and maintenance of ERP systems.

**Keywords:** ERP systems' usability problems; topic modeling; usability evaluation; heuristic evaluation; usability testing; cognitive walkthrough

## 1. Introduction

According to current research in Saudi Arabia, the software as a service (SaaS) market is projected to grow around 16.8% from 2018 to 2024. Following the vision 2030, the focus is upon the growth of the non-oil sector; one of these is the Information Technology (IT) industry. The commercial enterprises, and industrial and manufacturing end-users, currently own the largest revenue share in SaaS in Saudi Arabia [1]. According to the newest worldwide statistics, the Enterprise Resource Planning (ERP) market is projected during the forecast period 2020 to 2024 to grow at a CAGR (compound annual growth rate) of 9% with a revenue of USD 19.52 billion [2]. ERP is an efficient way of automating and integrating processes all over the organization [3]. An ERP system is provided by a cloud service provider in the SaaS model [4]. The ERP systems are extensively used to manage the operation and processing of manufacturers and can systematically connect different departments of an organization [5]. Therefore, ERP is a highly demanded IT solution in several sectors growing in any country, such as hospitality, healthcare, banking, and manufacturing. Furthermore, ERP systems are essential for the survival of highly competitive business environments to accelerate managerial decisions [6].

### 1.1. Usability of ERP Systems

According to the usability definition [7], the interface is designed to support the end-users in completing their tasks effortlessly in a particular context, and so they feel satisfied after using the system. Moreover, usability ensures that the interface is safe to use, provides utility, and is easy to learn and remember by the end-users [8]. Thus, usability incorporates specific quality attributes in the user interface that make it easy to use. In general, the benefit of usability is composed of three components: a rise in productivity, reduction in costs, and higher competitiveness [9].

As many organizations and sectors worldwide have adopted ERP solutions for an adequate flow of their processes, users face usability and user experience issues and challenges in using such systems [3]. The usability features are one of the most crucial elements that directly contribute to the success or failure of any software system. In addition, usability highly affects the project's acceptance by the end-users [10]. The usability problems are a barrier to users achieving their goals in the desired context of use [10,11]. The usability may lead to large-scale project failure, but it also interferes with the productivity of the user's workgroup. Therefore, projects need to spend at least 10% of their budget on usability to increase their effectiveness by 100% on sales, 161% on user productivity, and 202% on specific target features [12].

Many previous research papers highlighting the ERP usability issues focus on the area of information systems [13–15]. These contributions are primarily related to the adaptation of ERP systems by the users of corporates. However, it is essential to explore user interaction with ERP systems' interface usability issues. Identifying ERP systems' user interaction usability problems can help avoid such issues when implementing ERP systems. Furthermore, it is necessary to investigate the most effective usability evaluation methods in order to explore fully the ERP system usability issues. Consequently, that can help the experts to perform a quick usability test of ERP systems. There are several usability studies that have been performed to find ERP interface usability problems. Mainly these studies highlighted the usability issues of navigation structure [16], efficiency problems [3,11,17,18], lack of effectiveness [18], limited utility [18], interface complexity [3,11,16,18,19], inconsistency [20], lack of help [3,16,20], missing details [20], massive menu structure [16], and user experience (UX) problems such as exciting, dull, fun, appealing, and interesting [16], hard to search [17,19], memorability [17], unclear error messages [3,11], and vague feedback [3]. These ERP usability problems can affect the user's performance and professional growth. The experienced users learned to use their ERP system with practice; however, the inexperienced users still faced similar problems that caused performance degradation. Furthermore, the small and medium-sized organizations that cannot afford experts are more vulnerable to the adverse effects of usability issues [20]. The previous studies reported such problems individually; however, combining them in one document can benefit the experts as a quick guide related to ERP usability problems.

### 1.2. Objectives and Contributions

Usable ERP systems tend to reduce the training cost, usage frustration among the users, time to complete a task, and successful completion of projects. If the usability experts, interface designers, and developers are provided with comprehensive details of potential ERP problems and recommendations, it can help them to avoid these problems in designing and developing the ERP system. In addition, this data may help companies considering the usability evaluation of existing ERP systems to make improvement plans. For this purpose, we have proposed an approach for identifying ERP usability issues and developing usability improvement recommendations with the following objectives.

- Study ERP usability problems in the previous research papers and collect them for further analysis.

- Applying a machine learning approach to aggregate the major ERP system's usability issues found by the users. Consequently, utilize the results for developing a precise checklist related to ERP user–system interaction problems.
- Validate the reliability of the proposed ERP usability problem checklist by performing usability evaluations of existing ERP systems according to the proposed checklist.
- Develop the recommendations for the stakeholders to avoid frequent usability issues and report them to the community.

This paper proposed a state of the art approach to identifying the main usability issues in ERP systems using the machine learning technique of topic modeling based on previously reported usability evaluation results. Consequently, a precise list of frequent usability issues is developed based on the findings of the topic modeling in the previous step. ERP systems are evaluated using the methods of heuristic evaluation, cognitive walkthrough, and usability testing to validate the proposed ERP usability checklist. The output is used to develop the recommendations of developing usability interfaces of ERP systems.

For the remainder of this paper, Section 2 describes the previous contributions in this area. Section 3 explains our research methodology and usability evaluation, and usability recommendations. Finally, we close this paper with a discussion of our results and conclusion in Section 4.

## 2. Literature Review

This Section discusses the previous research on ERP systems concerning usability evaluation methods, findings, and contributions.

Many studies have reported usability evaluation studies of ERP systems. One of the most reported methods of ERP system evaluation is the usability testing method. The Dynamic Task Map with information (DTMi) [18] is an interactive interface to support ERP system tasks and is evaluated by usability testing. Additionally, many studies [3,18,21] assessed the ERP systems by using usability testing methods. These evaluations involved the actual users and ERP system experts in laboratory settings. The studies in [22,23] reported the survey findings that included the questions primarily related to the ease of use of ERP systems. Likewise, experiments were conducted to compare the usability of different ERP interfaces, such as in [17], where the authors evaluated the navigation and association support interfaces in SAP. In [16], an exploratory experiment study was conducted to compare the SAP with association map interfaces. Interviews were held in the study [11], beginning with an aim to address the problem of little documentation of the usability problems faced by users while using the ERP systems. Additionally, most of the studies mentioned above had conducted the initial literature review of the previous research to identify the kinds of ERP usability problems.

Mari-Klara and Wendy [3] conducted usability studies to develop the usability evaluation criteria of ERP systems. First, a total of 20 usability problem instances (UPls) were extracted from a collected 53 raw responses. Next, UPIs with similar primary issues were combined into ten general usability problems (UP). The limitation of the study is that the explored UP are only based on the data collected from the usability study of one ERP system and performed by the three participants, i.e., a limited number of evaluators and a limited number of systems. In [24], Singh and Wesson proposed heuristics that are specific to ERP systems. They proposed five criteria: navigation, presentation, task support, learnability, and task support for the usability evaluation of ERP systems based on general usability heuristics and rules after studying the ERP usability issues. Then, they conducted the usability study to verify the proposed usability heuristics in identifying the potential usability problems with an ERP system. The study is limited in terms of its application with heuristic evaluation. Ruuhijärvi [25] evaluated the usability of ERP systems by applying three methods of the cognitive walkthrough, heuristic evaluation, and user testing. There were nine usability issues found; out of these, error prevention and recognition problems had the most significant effect on the usability of ERP systems. The study

was limited to only one evaluator who performed the heuristic evaluation and a cognitive walkthrough. Additionally, usability testing was conducted from a remote location that may have caused some valuable observations to be omitted. Wüllerich and Dobhan [26] developed an end-user-based model for evaluating the usability of mobile ERP applications. They performed the usability evaluation on two ERP systems using the usability criteria of effectiveness, errors, efficiency, satisfaction, memorability, learnability, cognitive load, navigation, and presentation. The evaluation was conducted online and limited to the mobile ERP interface, and all the data were collected through questionnaires.

The previous research provides good data sources related to ERP system usability problems. However, these data need to be analyzed effectively to make them easy to use for the stakeholder beneficiaries of the ERP system in designing, developing, and improving the ERP user interfaces. Furthermore, none of the previous studies considered consolidating the issues related to ERP systems as a precise reusable checklist for designing ERP systems. This paper fills this gap and proposes a unique method of developing usability recommendations of ERP systems based on a machine learning technique.

### 3. Materials and Methods

This paper has presented and evaluated the conceptual approach of discovering usability problems in ERP systems. The primary outcome of this conceptual approach is a concise checklist of the common usability issues explored from the previously reported usability problems, which can facilitate the interested stakeholders in the quick discovery of them in ERP systems.

Figure 1 presents different components of our approach to develop usability recommendations for ERP systems. The recommendations help find usability issues while developing the ERP systems or existing ERP systems as a quick reference. The first module utilizes the current contributions and resources reporting the existing usability problems of ERP systems. A collection of reported ERP usability problems in textual form is the output of this component. An input of the second component is the usability problems data in a textual format, which is preprocessed to prepare it for topic modeling. A brief list of topics is analyzed to find a suitable title for the collection of similar problems. The third component is for the validation of the proposed topics. In this phase, the appropriate ERP usability evaluation method is first selected, and then the evaluator designs the evaluation of the available ERP system. This evaluation process is executed according to the design. Later, the reported problems are classified according to a concise checklist of topics related to the usability problem. This step can be repeated various times provided time and resources. Finally, the recommendations are developed for the checklist of topics related to the ERP system's usability problems. These recommendations are available for all interested stakeholders, whether they are interested in developing new ERP interfaces or evaluating existing interfaces to avoid the frequently occurring ERP usability problems.

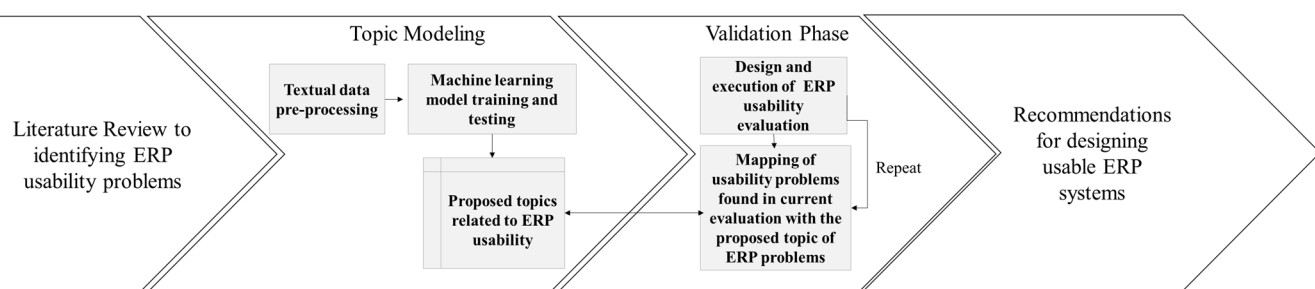

**Figure 1.** An approach for exploring usability problems in ERP systems.

In the following, we provide the details of each step of our approach for developing usability recommendations for ERP systems.

### 3.1. Exploring the List of Usability Problems from Previous Literature

First, we aimed to collect the data of previously reported usability problems of the ERP systems. Therefore, we searched the web to find relevant papers reporting ERP usability problems. For this purpose, the keywords used are +" ERP Usability issues", +" ERP Usability problems", and +" ERP system usability evaluation". Additionally, the keywords used to filter irrelevant research publications from our search are −" ERP critical success factors", −" adoption of ERP", −TAM, −" Technology acceptance model", −"acceptance model", "−adoption of ERP", "−TAM Technology acceptance model", and "−acceptance model". These keywords belong to ERP systems' adaptability models, which are out of the scope of our study. We selected the google scholar platform for searching purposes. We found 84 research papers as a result of searching the defined keywords. First, we chose the papers by reading their title, abstract, and keywords, and all the unrelated papers were discarded. As a result, we found 11 relevant articles reporting 89 ERP system usability problems. After removing the repeated problems, we were left with a total of 67 problems. All the problems listed in the research paper were stored in .csv format for further analysis. Next, we identified the clusters of similar ERP system usability issues reported in the previous literature. For this purpose, we adapted the topic modeling method as our data are not labeled, and we are required to identify the features related to this data. This helps in developing a concise checklist of ERP system usability problems. The following Section discusses the details of topic modeling on ERP usability problems' textual data.

### 3.2. Topic Modeling for Identification of Usability Issues in ERP Systems

Topic modeling is a clustering-based machine learning approach where text analysis is performed without the initial labeling of data. The topic model provides an automated procedure of coding a large chunk of text into a set of significantly meaningful coding categories that are called "topics" [27]. At the beginning, the researchers specify the number of topics to the algorithms to find in the provided text. The program is developed to identify the specified number of topics in the input text. The output is the probabilities of words being used in the topic and the distribution of those topics across the corpus of texts.

Latent Dirichlet allocation (LDA), first introduced by Blei et al. [27], is based on a generative probabilistic model of a corpus. This method uses the idea of distributing different documents as random mixtures over latent topics, and each topic is categorized by its distribution over words. The output consists of the words list in the groups of topics that provide the context of use [28]. The words with a higher probability provide the title of that topic. Albalawi et al. [29] compared the methods of semantic analysis, LDA, non-negative matrix factorization, random projection, and principal component analysis on short textual social data to investigate their benefits in detecting important topics. The results showed that the LDA defined the best and clearest meanings of Facebook conversation data compared to the other methods. Moreover, the study compared the outcomes of various topic modeling methods reported in the previous literature and identified the LDA as the most popular and highly studied model in many domains. It is also available in many toolkits such as machine learning toolkits of MALLET (https://mimno.github.io/Mallet/, accessed on 23 December 2021), Genism (https://radimrehurek.com/gensim/, accessed on 5 February 2022), and Stanford TM toolbox (TMT) (https://downloads.cs.stanford.edu/nlp/software/tmt/tmt-0.4/, accessed on 5 February 2022).

Therefore, in this paper, we have applied LDA over the collection of the list of problems reported by the ERP system users in the previous research. The reason for applying topic modeling on the collection of ERP usability problem data is to identify the critical and recurrent usability issues in interfaces. Consequently, the developers can use them as a reference and avoid them during the design and development activities.

As a first step in applying LDA, it is vital to calculate the optimal number of topics for the input textual dataset. For this purpose, we have calculated the Jaccard similarity

and coherence. Firstly, the Jaccard similarity coefficient [30] is applied to measure the similarity and diversity of sample textual input. Jaccard similarity coefficient is

$$J(A,B) = \frac{(A \cap B)}{(A \cup B)} \tag{1}$$

In Equation (1), *A* and *B* are two sets/topics; the common words in *A* and *B* are divided by all words in *A* and *B*. Furthermore, the measure of coherence is used to find the degree of semantic similarity between a single topic and high-scoring words in the topic [31].

Figure 2 presents six topics that are optimal for the input dataset of usability issues. The ideal number of topics is based on maximizing coherence and minimizing the topic overlap based on Jaccard similarity.

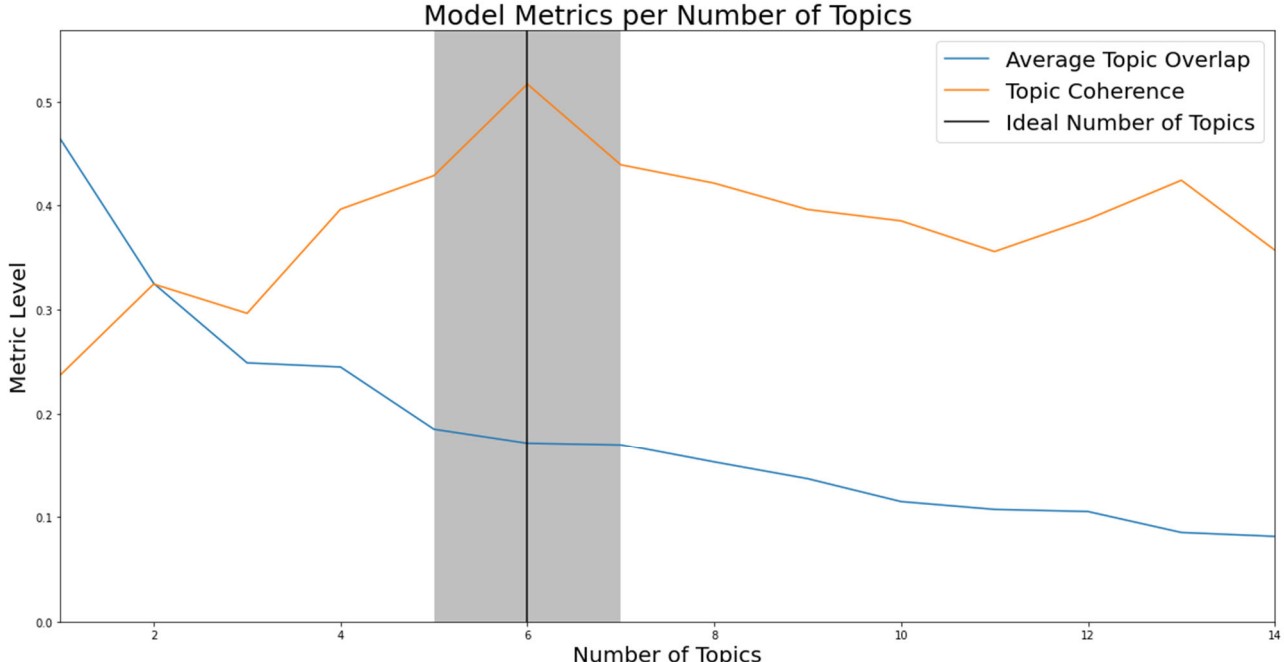

**Figure 2.** The measurements of an optimal number of topics of usability issues from the input dataset.

LDA is applied to the collection of usability problem data by following the various steps of data loading, cleaning, removing punctuation/lower casing, exploratory analysis, converting text to LDA analysis readable format, LDA model training, and model analysis and visualization of the results. The data is loaded from .csv format file. The data are cleaned by selecting the appropriate column of usability-related issues. Then, phrase modeling is performed by applying bigram and trigram modeling and making lemmatization. Furthermore, the special characters and stop words are removed, and data are converted to lower case letters. The initial preprocessing steps are validated by applying the exploratory analysis technique, and an expert visually validates the prepared data in readiness for further analysis. The preprocessing of data is continued by removing the steps words in the English language and some other undesired words. The input of the LDA model is dictionary and corpus, which are created by using the preprocessed data. The prepared data and the optimal number of topics are fed into the LDA model for training purposes. The model is implemented using $(LdaMulticore())$ method that allows running the model parallel using all CPU cores to speed up the processing [32,33]. The calculated topics related to usability problem are presented in Table 1. The words under each topic and the references of their data source are presented in the table. The chosen words belonging to each topic are highlighted in bold in the second column of top keywords.

**Table 1.** LDA calculated topics and their related words and the suggested titles for each topic by an expert.

| Predicted Topics | Top Keywords | References of Data Source | Experts Suggested Title |
|---|---|---|---|
| Topic 1 | "system", "**sap**", "menu", "**transaction**","**user**","location", "search", "**interface**","**easy**", "**difficult**" | [3,17–19,21–24] | Lack of feedback and complex interface |
| Topic 2 | "**user**", "entry", "**unclear**", "**system**", "field", "**rule**", "**information**", "**unhelpful**", "context", "require" | [3,11,18,23] | Unstructured data and information overload |
| Topic 3 | "user","**transaction**", "**find**", "system", "**step**", "**menu**", "**difficult**", "task", "**problem**", "**next**" | [11,17–19,21,24] | Difficulty in searching and finding desired item/information in interface and error handling |
| Topic 4 | "**specific**", "sap", "**perform**", "**task**", "**system**", "unclear", "login", "rule", "**report**", "**desire**" | [11,21,24] | Interface customization problems |
| Topic 5 | "**miss**", "**list**", "sap", "navigation", "**obvious**", "basic", "**selection**", "easy", "**transaction**", "feature" | [3,18,21,23,24] | Missing data and information |
| Topic 6 | "**user**", "function", "system", "**find**","sort", "**unclear**" | [3,18,19,22,23] | Contextual data and information organization issues |

The results presented in Figure 3 are then visualized for further analysis. The visualization of the results is available in the form of LDAvis proposed in [34]; it has two parts. The left part presents a global view of the topic model. It shows the frequency of occurrence of particular words related to usability problems and the relationship of topics related to usability problems. The right panel depicts the horizontal bar chart whose bars represent the individual terms that are the most useful for interpreting the currently selected topic related to the usability problem on the left and tells the meaning of each topic related to usability issues. Figure 3a presents the global view and frequency and the relationship of particular words in Topic 1 to usability problems; furthermore, Figure 3b–f represents the frequency and the relationship of particular words in Topics 2–6.

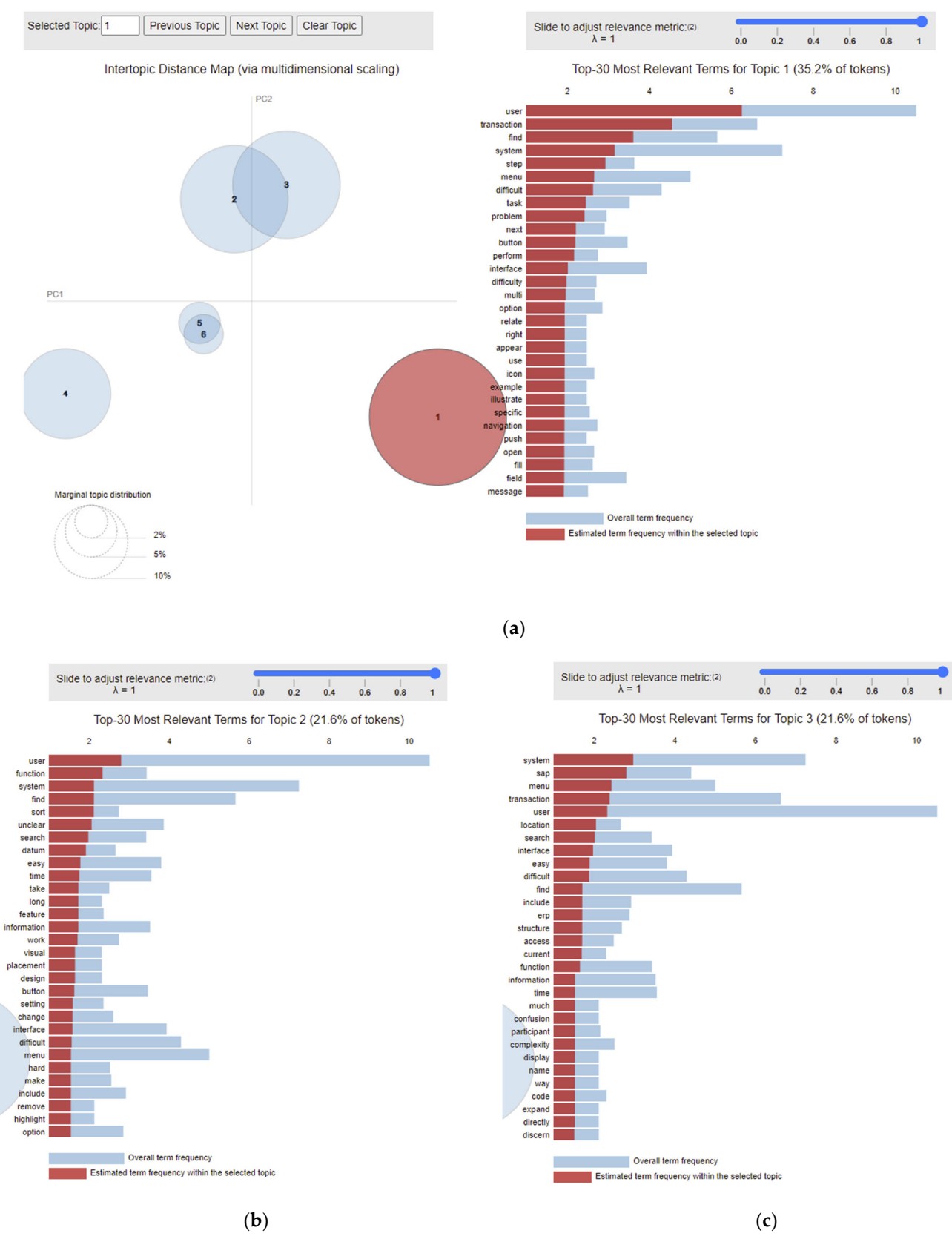

(**a**)

(**b**)

(**c**)

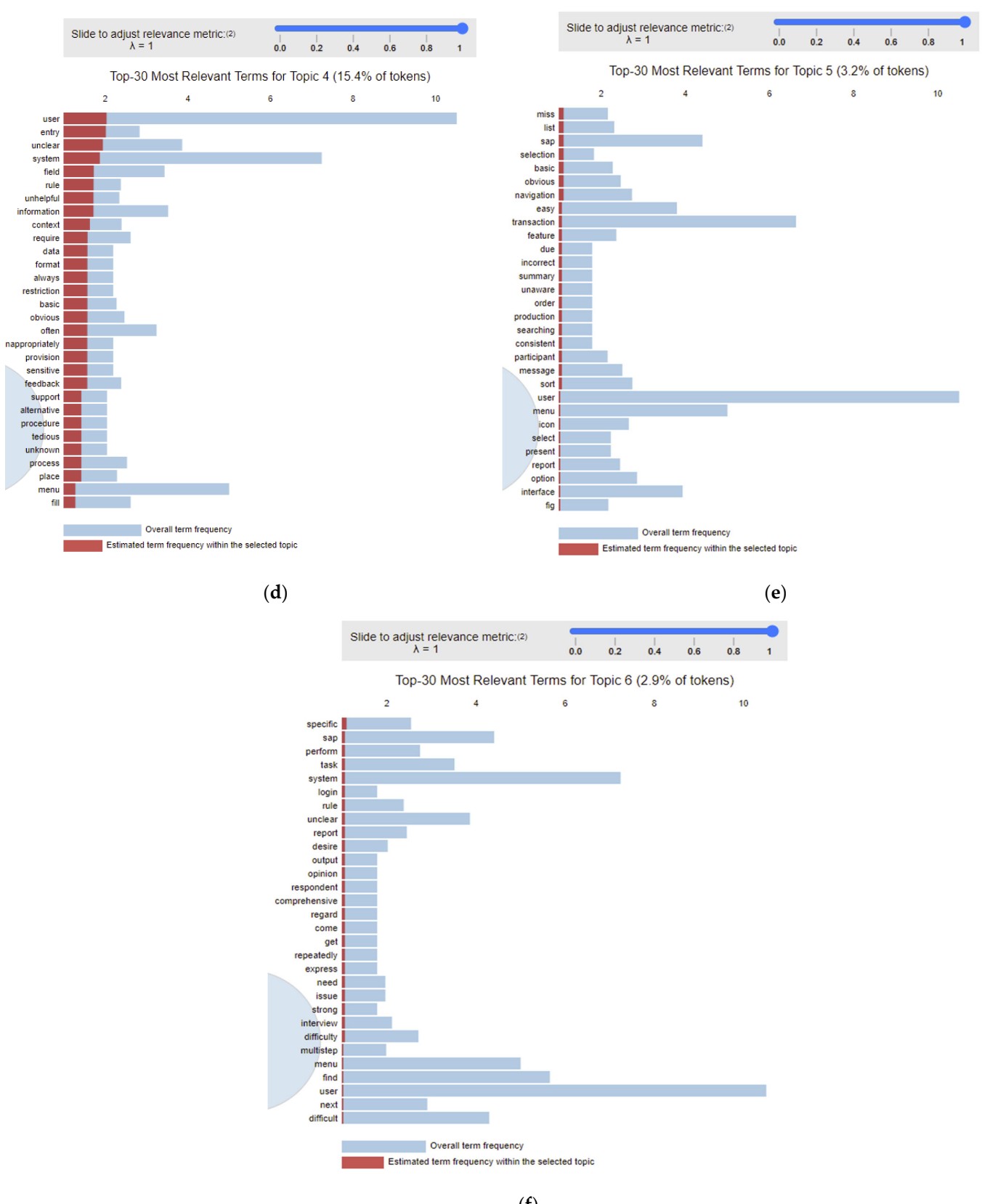

**Figure 3.** LDAvis presentation of LDA topic modeling of the ERP systems usability issues in (**a**) Topic 1, (**b**) Topic 2, (**c**) Topic 3, (**d**) Topic 4, (**e**) Topic 5 and (**f**) Topic 6.

After calculating the topics by applying the LDA method for topic modeling of usability problems of ERP systems. The method of (get_document_topics()) was used to obtain the topic distribution for the given problems reported in the previous research papers. This gave us a tag for each reported ERP usability problem document in the previous research papers. An expert manually investigated the reported problems under each topic to suggest a general title of that topic shown in the last column of Table 1.

### 3.3. Case Studies of Evaluating ERP Systems

The usability evaluation method consists of well-defined activities for data collection related to end-user interaction with the user interface that measure how a specific feature of this software product contributes to achieving a certain degree of usability [35]. The usability evaluation can be classified as with the users' and without the users' participation by the usability experts, analytics tools, and predictive models [8]. Usability inspection refers to popular usability evaluation methods involving the usability expert instead of the actual users [36]. The heuristic evaluation and cognitive walkthrough are the types of usability inspection methods. Usability testing is performed by involving the system's real users; therefore, it is classified as with users' evaluation method.

To validate the reliability of the proposed list of main usability problems, we have conducted the usability evaluation of the ERP system in Saudi Arabia's software firm. We first selected ERP systems' suitable usability evaluation methods in the reported case studies. Then, we designed the usability evaluation setup of the selected ERP system. Later, after the execution of usability evaluation of ERP systems, we mapped the reported ERP system usability problems with the topics in Table 1 (see Section 3.2). This study evaluated the ERP system using the three usability evaluation methods of heuristic evaluation, cognitive walkthrough, and usability testing to find the usability problems in a private ERP-based software company in Saudi Arabia.

### 3.3.1. Study 1: Customized Heuristic Evaluation Study

Heuristic evaluation is performed by an individual usability expert who completes the interface evaluation alone, and they are allowed to communicate and have their findings aggregated [37]. The evaluation process consists of three stages: briefing session, evaluation period, and debriefing session [8]. Ten usability heuristics (proposed by Nielsen and his colleagues [25]) are provided to the experts for interface evaluation. However, the usability heuristics can be customized according to the user interface type [38].

This study aims to employ customized heuristics in evaluation to find the maximum ERP usability problems. The evaluation was conducted to locate problems utilizing the criteria identified in the previous studies and report unique challenges users can face while interacting with the interface. Although heuristics developed by Jacob Nielsen [8] can be applied to various products, many-core heuristics are too general to evaluate products based on emerging technologies, including ERP systems. Therefore, the heuristics must be customized according to the ERP systems' interface requirements. Many studies have suggested the customization process of usability heuristics according to the type of products and context of use [39–41].

#### Procedure

The first step of our evaluation process was the customization of the heuristics for ERP systems. The initial version of heuristics was developed and sent to 4 usability experts for validation purposes. Then, we updated the customized heuristics of ERP interface evaluation for this study according to their feedback. Table 2 presents the customized usability heuristics that we used in our study. The four male and female experts, experienced in usability and ERP systems, took part in the evaluation. They were expected to go through the form for reporting the evaluation and sign the consent form. The evaluation forms consist of an overview of system details, the rating criteria of each heuristic, and the

overview of the heuristic evaluation process. Secondly, before the session, they reviewed the customized HE checklist shown in Table 2. We ensured that the participants understood Jakob Nielsen 's four-step scale ranking procedure.

**Table 2.** The customized heuristic of ERP systems.

| Name | Heuristics |
|---|---|
| Customization | • The ease with which the system can be configured to a particular business type.<br>• The capability of the system to support user-level customization.<br>• The ability of an interface to be configured without affecting the system's underlying business logic. |
| Task support | • The ability of a system to improve user productivity.<br>• The various functions of the system can be identified by exploration.<br>• The user will be able to understand the task flow. |
| Privacy | • The system does not store any confidential information.<br>• The specific authentication procedures are offered to save the access of protected or confidential areas. |
| Visibility of system status | • Always keep users informed about what is going on.<br>• Provide appropriate feedback within a reasonable time.<br>• The clarity in feedback. |
| Alignment between system and the practical environment | • Speak the user's language with words, phrases, and concepts that are familiar to the user, rather than system-oriented terms.<br>• Follow real-world conventions, making information appear in a natural and logical order. |
| Control and freedom of the user | • Users should provide the option to reverse the actions as they often choose system functions by mistake.<br>• Provide a clearly marked "out" to leave an unwanted state without having to go through an extended dialogue.<br>• The interfaces should support undo and redo actions.<br>• The system prevents the user from making the error. |
| Error Prevention | • There are default values contained in the fields.<br>• The system warns users if they are about to make a potentially serious error.<br>• Make objects, actions, and options visible. |
| Recognition instead of recall | • Users should not have to remember information from one part of the dialogue to another.<br>• Instructions of the system's use should be visible or easily retrievable whenever appropriate.<br>• Users find it easy to access the desired information. |
| Flexibility and efficiency of use | • Users can easily access the system's functionalities.<br>• There must be a similarity between a queried item and the expected result.<br>• Accessing the user interface accommodates both experienced and inexperienced users.<br>• Dialogues should not contain irrelevant or rarely needed information. |
| Aesthetic and minimalist | • Every extra unit of information in a dialogue competes with the relevant units of information and diminishes their relative visibility.<br>• It may be necessary to provide help and documentation, even though it is better to use the system without documentation. |
| Documentation and help | • Help information should be easy to find, and searching should be easy to carry out and simple.<br>• Errors should be expressed in plain language (no codes). |
| Help users recognize, diagnose, and recover from errors | • The Error message should precisely indicate the problem.<br>• The dialog should constructively suggest a solution to recover from the error.<br>• A user can learn how to use the system without a long introduction.<br>• The user will be aware of the function to recover from the errors. |

Results

Table 3 presents many problems found in user interaction with the interface related to the checklist of the usability problems highlighted in Table 1. The associated concepts between the two columns are shown in bold words in the last column. The evaluator noticed missing help documentation in the user interface while using "help" feature. As well as that, the only available option was to send a defect report. Here, the user needs help according to the context of use, which is related to topic 6 (see Table 1). The experts found that some pages have more than one icon for refreshing the page while browsing the system; therefore, it may confuse the users. Furthermore, the icon title was not in the same language on every page, as some pages are in English, and others in Arabic. The interface required adequate structure to present information, as related to topic 6.

**Table 3.** The usability problems reported by participants in the heuristic evaluation of ERP systems.

| S.N | The Found Usability Problems | Usability Problem under the Topics in Table 1 |
|---|---|---|
| 1 | The interface is a mixture of two languages (Arabic and English). | 6 (Contextual data and **information organization issues**) |
| 2 | The help option functionality is lacking help. | 6 (**Contextual data** and information organization issues) |
| 3 | There is no help and system-related documentation available. | 6 (**Contextual data** and information organization issues) |
| 4 | Many actions have no clear feedback. | 1 (**Lack of feedback** and complex interface) |
| 5 | There is more than one refresh button on the same page. | 2 (Unstructured data and **information overload**) |
| 6 | The number of dialogues related to various functions is limited. | 2 (**Unstructured data** and information overload) |
| 7 | The flow of functions is not clear. | 3 (**Difficulty in searching and finding desired item/information in interface** and error handling ) |
| 8 | The system does not show any alert if something crucial happens. | 3 (Difficulty in searching and finding desired item/information in interface and **error handling**) |

Moreover, experts noticed that the user must understand the system flow before using it. This means that it can impact the learnability of the system, and the various operations of the system cannot be identified by exploration. As a result, it affects the users' productivity as this problem is associated with Topic 3. Similarly, selecting a suitable ERP solution for a wide organizational application can be challenging because of many determining factors affecting the selection criteria related to business development.

3.3.2. Study 2: Cognitive Walkthrough

Cognitive walkthrough [42] is a type of inspection method that is developed to evaluate the ease of learning of a user interface. The process of cognitive walkthrough consists of five steps: prerequisite is to identify the user characteristics, context, tasks, and description mock-up or prototype. Then, during the session, the experts and designer sit together to analyze; they need to answer three questions. In the end, the recorded information is compiled. Finally, identified problems are required to be fixed [8]. This method is focused on identifying specific users' problems in using the provided interface.

The study was conducted to validate and evaluate the ERP system from the users' perspective, leading them to the correct action to achieve their goals. Table 4 shows a list of the tasks developed for this study:

**Table 4.** Tasks designed for cognitive walkthrough study.

| Tasks | List of Tasks for the Cognitive Walkthrough Study |
|---|---|
| **Task 1** | Check the monthly salary sheets for last month. |
| **Task 2** | Change the payroll of Alexandra Adrian Allan Andrew to 8000. |
| **Task 3** | Add leave balances for employee Anne Boris Bell Gordon, which starts on 02/09/2019 until 09/09/2019. |
| **Task 4** | Add new allowance for employee Audrey Brandon Berry Harry with amount 900. |

Procedure

Four usability and ERP system experts participated in the cognitive walkthrough study. We selected one ERP system to evaluate in this study. The experts were provided with the four tasks presented in Table 4. The experts performed each task guided by the three following questions:

Q1: Will the correct action(s) be evident to the users? (The actions they know to achieve the task.)

Q2: Will the users find the proper control to be used for their next action, and it appears when needed?

Q3: Will the user receive sensible feedback that tells if they have made a correct or incorrect choice of action?

During the walkthrough, the experts recorded the critical information and usability issues reported in the results section.

Results

The concepts associated with the usability problems checklist are shown in bold words in the last column. Table 5 reports the usability problems found during the cognitive walkthrough evaluation of the ERP system. The different usability problems found in the ERP system are related to the topics listed in Table 1. The concepts that map the problems with the usability problems checklist are shown in bold words in the last column.

**Table 5.** The ERP usability problems found in the cognitive walkthrough study.

| Tasks | Problems | Usability Problem under the Topics in Table 1 |
|---|---|---|
| Task 1 | There are many options makings the users confused. | 2 (Unstructured data and **information overload**) |
| | Some of the titles are written in Arabic, and some of them are in English on the same interface. | 6 (Contextual data and **information organization issues**) |
| | When the user opens sub-tabs, it is confused with the primary tabs. | 3 (**Difficulty in searching and finding desired item/information in interface** and error handling) |
| Task 2 | A few tasks are restricted, and no permission to perform them. | 4 (**Interface customization problems**) |
| | The edit icon option does not allow editing anything. | 4 (**Interface customization problems**) |
| Task 3 | This task can be completed in one step, but currently, it is required to complete in multiple steps. | 3 (**Difficulty in searching and finding desired item/information in interface** and error handling) |
| | Categories and sub-categories are not clear. | 5 (**Missing data and information**) |
| | Labels' names are not clear. | 5 (**Missing data and information**) |
| Task 4 | There is no option available to select the employee's name first. | 4 (**Interface customization problems**) |
| | There is no button to click on the system to send the new amount for the system. | 5 (**Missing data and information**) |

In particular, the study shows that availing too many user options on the ERP user interface causes about 30% of the total ERP usability problems. For instance, too many options to the system user interface are available to select in the first task, confusing the user. Therefore, it is a type of "information overload" problem related to topic 2 (see Table 1). Moreover, the same interface is developed in two different languages of English and Arabic, which raises the misunderstanding among the users concerning the roles and consequences of various system functions. We found a similar issue in the heuristic evaluation study.

Furthermore, some basic functionality is missing that can facilitate the users to input new values into the system when required, which is a type of "missing information" related to topic 5 (see Table 1). Moreover, inadequate or unclear feature labeling may also cause problems to users. Additionally, it was found that the edit function could not perform the function as expected; thus, users were not allowed to amend or edit any information entered, which is the problem of "customization" and related to topic 4 (see Table 1).

### 3.3.3. Study 3: Usability Testing

Usability testing [43] is an interface evaluation method where products are tested in controlled laboratory settings involving real users. The users are provided with the predefined tasks to perform on the interface, and their data are recorded through direct observations, videos, audio, or questionnaires. Usability testing aims to find the product's usability and identify the interface problems.

We performed usability testing on the selected ERP system. In total, 7 participants, 3 male, and 4 females, with computer skills experience with expertise as advanced beginners, took part in the study. Advanced beginners understand causes and fix common errors, though they may not know precisely why the errors occurred and are self-sufficient primarily when performing tasks.

The users were required to perform the tasks presented in Table 4. The following measures were considered in usability testing studies based on users' tasks during the session.

1. Task success (0, 0.5, 1).
2. Error number (how many errors made by a user while performing a task).
3. Time spent using Keylogger.
4. User satisfaction (1–5).

Our expectations for the users were for them to identify quantitative and qualitative problems from their experience. Thus, compile a list of usability problems and recommendations after the test.

### Results

Table 6 presents the usability problems found by the users in ERP systems. Many usability problems belong to the proposed checklist of usability issues of ERP systems.

**Table 6.** Usability problems found during the usability testing evaluation.

| Task | Problems | Usability Problem under the Topics in Table 1 | Percentage of Problems in This Task |
|---|---|---|---|
| Task 1 | Bank report request control is hard to reach. | 3 (**Difficulty in searching and finding desired item/information in interface** and error handling) | 42.80% |
| | Labels of user interface controls are unclear. | 3 (**Difficulty in searching and finding desired item/information in interface** and error handling) | |
| | The system does not show a warning before any action. | 1 (**Lack of feedback** and complex interface) | |
| Task 2 | Some area of the interface was not visible due to its layout so users cannot find the information. | 5 (**Missing data and information**) | 14.20% |
| Task 3 | The information in the dropdown list was unclear. | 5 (**Missing data and information**) | 28.57% |
| | It takes a very long time to select time duration of leave balances for employee. | 1 (Lack of feedback and **complex interface**) | |
| Task 4 | Confirm step buttons was unclear on the interface. | 5 (**Missing data and information**) | 14.20% |

The Keylogger tool is used for logging the users' clicks to obtain the average time needed to complete the task in Table 7. We found the usability problems from the users' perspective in the usability testing. Most of the issues were related to "missing data and information," related to topic 5 (see Table 1). This issue is major as it causes the degradation of the users' performance. It is clear from the Keylogger tool data that users took a longer time to complete the tasks.

**Table 7.** Time calculated using the keylogger.

| Task | Time Spent |
|---|---|
| Task 1 | 4:09 min on average to reach the request step. |
| Task 2 | 2:11 min to enter the new payroll. |
| Task 3 | Entering the period of new leave balance was confusing with the previous leaves list, which takes 3:19 min on average to catch. |
| Task 4 | Users took a long time to confirm the amount, and some of them could not. |

In the next Section, we present the recommendations for designing user interfaces of ERP systems based on our findings and observations from the previous literature and usability evaluation studies.

### 3.4. Recommendations to Avoid the Usability Problems in ERP Systems

Based on our approach, six different topics are modeled to find the critical and recurrent usability issues in ERP systems. The three usability evaluation studies are performed to validate ERP usability issues. The results show that similar ERP usability problems occur while using the different ERP systems and conducting various usability evaluation studies. In the following, we propose recommendations to design usable ERP systems' user interfaces by avoiding the most common usability issues.

#### 3.4.1. Lack of Feedback and Complex Interface

The lack of feedback and a complex interface is the most occurred problem reported while using ERP systems. Ben Schneiderman suggested the eight golden design rules [44] to develop usable user interfaces applicable to most interactive systems. Similarly, Nielsen's ten heuristics [37], present an essential checklist of usable interfaces. While designing the ERP system interface, the designer should consider applying informative feedback to the user interface. This means to present implicit or explicit responses to every action of the user. Moreover, the feedback response can be different according to the intensity of the user's action. It can be more suitable if the change is shown explicitly to the users. It is

recommended to provide the appropriate feedback to the ERP users at a suitable time. The feedback should be designed by following the usable design method on each action of the users. Often, feedback messages should be presented to the user as a written or auditory notification. It can be implicit in the design, such as changing color or font.

In the direction of the complex user interface, users have found the structure of information arrangement to be complex. It is difficult for them to remember the path of information access, and the SAP interface is often crowded and messy. It is recommended to make the interface items visible to the users by providing the appropriate widgets that afford recognition to minimize user's memory load. Similarly, the design should discourage memorizing any command or path throughout the interface. Users should be given enough time to be trained for complex transactions on the interface. The interface is required to be efficient to use by reducing the number of steps for task completion.

### 3.4.2. Unstructured Data and Information Overload

In the previous research papers, many users highlighted the problems of missing information, too much information presented in an unstructured manner, hard-to-read information, and complex information structure. This can increase memory load and make it difficult to remember much information. It is recommended to make items and information visible to the users and avoid such commands that need to be remembered to perform the tasks. Additionally, the interface must be efficient to use [8], which allows the user to complete their tasks in a minimum number of steps. The design of the interface supports the concept of memorability. Therefore, the items are arranged in logical groups and presented as visual list options.

The problem of information overload can be avoided by preventing such information on the top that is rarely needed by prioritizing the requirements. Additionally, too much change in interface behavior that may surprise the users and annoy them should be avoided. Data entry should be made simple and guided for the users. The desired information should be easy to obtain to support the users in reaching their goals. In this regard, the interface can be supported by using the auto form filling method and constraining the users from entering the information in irrelevant formats.

### 3.4.3. Difficulty in Searching and Finding Desired Item/Information in Interface and Error Handling

One reported ERP usability problem is searching for desired interface items and information. Such problems may be due to information structure, which is a hurdle in the learnability of such a user interface. The users must be provided with sufficient time and training to learn such an interface. The searching can be made easy by combining the recognition and recall in the interface. The user can enter relevant letters and words; a browsable list with suggestions appears in the response. The ERP system should come with a reasonable amount of help documentation and tutorials that users can refer to so as to complete their tasks. In this regard, the designer should provide the users with an appropriate system usage guide. Moreover, the interface items should be easier to understand for the users. In this regard, the interface should speak the user's language with words, phrases, and familiar concepts and avoid system-oriented terms. It is also essential to develop the ERP user interface by following the standard to maintain the consistency that encourages users to learn efficiently.

The users must be empowered to recognize the interface errors, diagnose them, and recover for error prevention. The interface should constrain the users by greying the item where they try to access such controls that are not appropriate in that context. Many items can be available as a dropdown list; the data entry should not accept incorrect data formats. The error messages should be meaningful and provide the users with a guide to recover from errors.

### 3.4.4. Interface Customization Problems

Many ERP users report the interface customization problem. It is difficult for them to change any ERP settings, adapt them according to their wishes, and obtain the desired output in the preferred format. It is reported by the users that "finding specific functionality quickly within the system sometimes required an unreasonable amount of effort". The ERP systems must allow users to modify the interface settings and provide different layout options for the transactions outputs. The system should cater to both experienced and inexperienced users. The system should support the experienced users as they desire the sense that they are in charge of the interface. Therefore, the ERP system should support the options of several views, and users can add or remove any user interface object. According to [45], it is important to show users that customization is available by providing some labels and titles, making the customization easy, and allowing users to go back to the previous selection. A method of visual command [46] suggests that the user activity must be followed by interface feedback and present its description.

### 3.4.5. Missing Data and Information

Several users have complained about missing data and information of transactions and that the navigation and selection are not obvious. This problem can be avoided by providing a consistent interface and feedback to the users. According to Schneiderman's rule, the action should be presented in sequence to support users' tasks. The sequence of action should be organized into groups, and informative feedback should be provided after completing an action to provide the users with the subsequent satisfaction of accomplishment. If the designer keeps their users informed about the status of every step, then the users cannot get the feel of missing data. While presenting widgets on the user interface, we should keep the principle of visibility in mind by making them visible to users. ERP systems' interfaces should be developed based on the usability principle of affordance, which ensures that users should know how to use them.

### 3.4.6. Contextual Data and Information Organization Issues

One of the problems highlighted by the ERP systems' users is knowing "what to do on the current interface". Sometimes this happens due to information overload, which makes it challenging to obtain the objective of the current interface. It is recommended to make the purpose of the current interface more obvious, highlight the data by color coding, make frequently used menu items prominent and valuable, and the grouping of menu and titles should be meaningful. Nielsen and Schneiderman highlighted the importance of consistency and standards for implementing user interfaces for developing menus, help screens, color schemes, layout, and fonts throughout the system.

## 4. Discussion and Conclusions

This paper presents a state of the art approach to developing a precise list of the recurrent usability issues found in ERP systems. First, we have explored the existing literature on ERP system usability problems reported in usability studies. Second, the reported data is analyzed using the topic modeling approach of LDA to identify featured usability problems. The LDA algorithm output of each previously reported usability problem is mapped under each of the six topics. An expert manually analyzes these data, and each topic is titled as: (1) "lack of feedback and complex interface", (2) "unstructured data and information overload", (3) "difficulty in searching and finding desired item/information in interface and error handling", (4) "interface customization problems", (5) "missing data and information", and (6) "contextual data and information organization issues". Third, these topics are validated in the usability studies of ERP systems using three usability evaluation methods and, the results show the usability problems are associated with these topics. A total of 27 usability problems were found in three studies. The most common

usability problems are "difficulty searching and finding desired item/information in interface and error handling" and "missing data and information". Moreover, the second most frequently reported problem is related to "unstructured data and information overload". In the usability studies, we have found ERP system usability problems related to all the topics. Lastly, we have developed the recommendations for ERP usability-related problems primarily based on Nielsen's ten heuristics, Schneiderman's eight golden rules, and usability principles. The outcome of our paper is a precise list of ERP usability issues and recommendations to avoid them that can be used by the stakeholders in designing, developing, and evaluating ERP systems.

The previous research was available with many different types of ERP system usability problems reported in numerous documents. This paper has combined these data and presented them using a unique approach by summarizing them optimally. Therefore, our study facilitated all interested stakeholders in improving their ERP systems based on previously reported experiences of usability problems; such a contribution is lacking in the previous research. The previous study [3,24] developed the heuristics customization approach for the ERP systems usability evaluation criteria. In contrast, this study developed the criteria using a machine learning method from the collection of previously reported problems. Therefore, the results are justified, as the discovered topics cover all the previously developed usability criteria of ERP systems. The findings of this study are validated through usability evaluation of ERP systems in three different studies. However, the outcome can be generalized to various ERP systems.

Recently, the results of usability evaluation studies reported problems with mobile, cloud, and SaaS interfaces. In [26], the authors evaluated two mobile ERP interfaces and reported the aspect of navigation and the presentation of the content causing a cognitive workload. This aspect is highlighted under the proposed recommendation of "missing data and information" in our contribution. The usability problems related to mobile ERP systems such as a complex interface, being hard to remember, trouble in finding desired information, a lack of help and documentation, the context of use and guidance, and information overload are reported in [47], which are precisely mapped with the outcomes of our study. In [48], the authors highlighted the usability problems of customization, information provision and feedback, the complexity of multistep tasks, data entry, visual design, searching, and navigation over lists while comparing the SAP S/4HANA and Oracle Cloud ERP cloud-based ERP systems. The usability problems emphasized in the previous studies related to mobile interfaces and cloud-based ERP interfaces are similar to the problems we highlighted and used to develop the recommendations. Therefore, we observed the effectiveness of our usability recommendations in a more comprehensive range of ERP systems.

We have based our study on the data previously reported on the ERP usability problems from the usability studies; therefore, our model is established on published data. The developed usability criteria are grounded and evaluated on ERP systems' interfaces. In the future, the proposed usability recommendations can be customized for developing the emerging user interfaces for ERP systems, including mobile- and IoT (internet of things)-based ERP systems. This research can help the companies improve their ERP systems' user interfaces that might consequently help increase productivity and reduce the cost of the companies adopting the ERP systems.

**Author Contributions:** Conceptualization, A.A.; data curation, D.A.; formal analysis, D.A.; funding acquisition, M.A.A.; investigation, A.A. and D.A.; methodology, A.A.; project administration, M.A.A.; resources, D.A.; supervision, M.A.A.; validation, A.A.; writing—original draft, A.A.; writing—review and editing, A.A. All authors have read and agreed to the published version of the manuscript.

**Funding:** This research received no external funding.

**Institutional Review Board Statement:** Not applicable.

**Informed Consent Statement:** Not applicable.

**Data Availability Statement:** Not applicable.

**Acknowledgments:** The authors gratefully acknowledge King Faisal University, Al-Ahsa, Saudi Arabia, for providing the resources to support this research contribution.

**Conflicts of Interest:** The authors declare no conflict of interest.

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
