# Peer review of "A Comprehensive Approach of Exploring Usability Problems in Enterprise Resource Planning Systems"

_applsci, doi:10.3390/app12052293_

Round 1
Reviewer 1 Report
I believe the current research is well-written dealing with relevant and important issues. I enjoyed reading the paper a lot. The research findings can provide insightful implications about usability issues of ERP systems during the development and maintenance of the ERP systems.
I only have some minor suggestions.
First, please try to consider changing the title. You may use Comprehensive instead of Automated. Or you may consider using a machine learning technique.
Second, please add lines in table for Table 1, Table 2, Table 3, Table 4, Table 6, Table of line 480, Table 7 and, Table 8.
Third, add table number and title at line 480. And adjust the following table numbers accordingly.
Lastly, please consider factors affecting intention to adopt Cloud-based ERP on the top of the mobile ERP system interfaces.
Thank you very much for the authors' efforts.
Author Response
Thank you so much for your review and constructive feedback. The completed review response report is attached as a word document. The revised article is uploaded in two forms i.e., clean and track changes

Reviewer 2 Report
The proposed paper focuses on an interesting topic, proposing a novel approach. Nevertheless, some issues should be addressed: The paper should be reorganized in order to increase its comprehensibility. The scientific novelty of the paper should be remarked. Explain, please, how the proposed approach is superior to other ones previously reported in the literature. Why the LDA was selected among other techniques? This selection should be justified. Could the proposed approach be extended for solving other problems? The paper is excessively extended. Some changes should be considered for reducing its extension. For example: (i) Introduction and Section 1 could be integrated into a single section; (ii) Section 4 could be reduced and integrated into Section 3; and (iii) information shown in Table 3, could be summarized. English grammar should be checked.Author Response
Thank you so much for your review and constructive feedback. The completed review response report is attached as a word document. The revised article is uploaded in two forms i.e., clean and track changes

Reviewer 3 Report
The authors present a proposal for an analysis of the ERP usability problem, based on preliminary studies, with the application of the machine learning technique (LDA).
The case studies were applied in software firms (SaaS) in Saudi Arabia.
They present recommendations to address the main issues associated with the development of an ERP system.
Most of the recommendations presented are based on the work of Nielsen and Shneiderman.
In table 1, it is desirable to clarify which words belong to the respective topics.
Figure 3: hard to see, need to improve presentation.
In tables 3, 6 and 7, it is desirable to clarify which items in column 2 belong to those in column 1.
Table 7 is duplicated.
Several times Nielsen's name is misspelled.
Bibliographic reference 23 is incomplete.
I consider as interesting contributions the analysis of previous studies of ERP systems with the application of the machine learning technique (LDA).
To validate the proposal, I think the authors need to study the adherence and effectiveness of the methodology and recommendations presented, considering a wider range of ERP systems.
Author Response

(The authors gave the same response as above.)

Round 2
Reviewer 2 Report
I do consider all the previous concerns have been satisfied in the revised manuscript.